# An osmium-peroxo complex for photoactive therapy of hypoxic tumors

Nong Lu[1,4], Zhihong Deng [2,4], Jing Gao[2,3], Chao Liang[1], Haiping Xia [2✉] & Pingyu Zhang [1✉]

The limited therapeutic effect on hypoxic and refractory solid tumors has hindered the practical application of photodynamic therapy. Herein, we report our investigation of an osmium-peroxo complex (**Os2**), which is inactive in the dark, but can release a peroxo ligand $O_2^{\bullet-}$ upon light irradiation even in the absence of oxygen, and is transformed into a cytotoxic osmium complex (**Os1**). **Os1** is cytotoxic in the presence or absence of irradiation in hypoxic tumors, behaving as a chemotherapeutic drug. At the same time, the light-activated **Os2** induces photocatalytic oxidation of endogenous 1,4-dihydronicotinamide adenine dinucleotide in living cancer cells, leading to ferroptosis, which is mediated by glutathione degradation, lipid peroxide accumulation and down-regulation of glutathione peroxidase 4. In vivo studies have confirmed that the **Os2** can effectively inhibit the growth of solid hypoxic tumors in mice. A promising strategy is proposed for the treatment of hypoxic tumors with metal-based drugs.

[1] College of Chemistry and Environmental Engineering, Shenzhen University, Shenzhen 518060, China. [2] Shenzhen Grubbs Institute, Department of Chemistry, Southern University of Science and Technology, Shenzhen 518055, China. [3] Center for Reproductive Medicine, the Third Affiliated Hospital of Sun Yat-sen University, Sun Yat-sen University, Guangzhou 510630, China. [4] These authors contributed equally: Nong Lu, Zhihong Deng. ✉email: hpxia@xmu.edu.cn; p.zhang6@szu.edu.cn

The hypoxic microenvironment of solid tumors caused by abnormal proliferation and vascularization of cells can greatly compromise the therapeutic effect of traditional photodynamic therapy (PDT)[1], which usually requires $O_2$ for the production under light activation of reactive oxygen species (ROS) by a photosensitizer[2–9]. Many innovative methods have recently been introduced to solve the problem of tumor hypoxia. Some additives such as perfluorocarbons, artificial red blood cells, and covalent organic frameworks[10–12], carry $O_2$ directly to the tumor site, and others utilize unique in situ features of the tumor microenvironment to generate $O_2$. These include $CaO_2$ and graphite phase-carbon nitride ($gC_3N_4$) carbonization[13,14] or high $H_2O_2$ concentrations in tumor cells resulting from $MnO_2$ or $H_2O_2$ decomposition[15,16], or even photocatalytic oxygen production by photosynthetic bacteria[17]. These nanomaterials drugs however are still not ideal and there are numerous possible adverse side effects[18]. In addition, in vivo studies have shown that enrichment of $O_2$ promotes proliferation and inhibits apoptosis of malignant cells, results which are not conducive to PDT[19]. Therefore, improvement of the efficacy of phototherapy for hypoxic tumors is necessary.

It has been reported that some metal complexes show excellent PDT effect toward hypoxic tumors[20–26]. Most of them do not depend on oxygen through traditional type II mechanism for PDT. For example, Sadler et al. studied that an iridium (III) photoredox catalysis can provide an oxygen-independent mechanism of action to combat hypoxic tumors. The iridium (III) complex photocatalytically oxidized 1,4-dihy-dronicotina-mide adenine dinucleotide (NADH), an important coenzyme in living cells, to generate NAD• radicals with a high turnover frequency, and synergistically photoreduced cytochrome C under hypoxia[20]. Chao et al. designed an iridium (III) complex that upon irradiation, produces free carbon radicals under hypoxia[26]. In addition, photoactive chemotherapy (PACT) has the potential to overcome the limitations imposed by hypoxia. In PACT, metal complexes can be photoactivated in a controlled manner to produce cytotoxic substances. Toxic gases such as $NO$[27] and $CO$[28], or ligand-centered cytotoxic substances can always be released by photoactivation[29–32]. The superoxide radical ($O_2^{\bullet-}$) is one of the most toxic ROS and has been identified as a most useful oxidant for cancer treatment and an adjuvant in synergistic chemotherapy[33,34]. Under the action of intracellular superoxide dismutase, $O_2^{\bullet-}$ can form $H_2O_2$ and $O_2$. The accumulated $H_2O_2$ is further transformed into a hydroxyl radical (OH•) with enhanced toxicity and reactivity, exacerbating oxidative damage of cancer cells and improving the anticancer effect[35]. The design of a photoactive compound that does not rely on $O_2$ but is photo-controlled to release ROS has significant application prospects in hypoxic tumor treatment.

In this work, we investigate an osmium-peroxo complex (**Os2**), whose structure is stable in the dark. However, upon photoactivation with 465 nm light, **Os2** releases $O_2^{\bullet-}$ even in severely hypoxic conditions (1% $O_2$), and is transformed into the cytotoxic **Os1** in a $Cl^-$ containing PBS solvent (Fig. 1a), thus maintaining useful photoactivation efficacy in hypoxia. **Os1** displays toxicity in both light and dark, contributing to a synergistic effect of chemotherapy and PACT of **Os2** under light irradiation. We also found that irradiation of **Os2** induces distinct ferroptosis mediated by GSH degradation, lipid peroxide accumulation and GPX4 down-regulation. Furthermore, **Os2** can photocatalytically oxidize endogenous NADH in living cancer cells, triggering ferroptosis. In vivo studies confirm that **Os2** effectively inhibits the growth of solid hypoxic tumors in mice. Thus this work is developing a process to release $O_2^{\bullet-}$ for the treatment of hypoxic tumor cells through the mechanism of ferroptosis.

## Results

**Photoactivation property.** The synthesis of the osmium-peroxo complex **Os2** was based on our previously reported method[36], and the characterization data can be found in the Supporting Information (SI, experimental section and Supplementary Figs. 1–3). **Os2** is highly stable in phosphate-buffered saline (PBS) pH = 7.4 in the dark at room temperature (rt) (Supplementary Fig. 4). However, with 465 nm light irradiation, the solution of **Os2** undergoes a gradual color change from brown to light pink (Fig. 1b, insets), which led us to investigate this transformation in detail. High-resolution mass spectrometry (HRMS) measurements were performed to identify the photolytic products, and a prominent peak was observed at $m/z = 1175.2351$ (Supplementary Fig. 5). The mass and isotope distribution in this ion correspond well to **Os1**. The formation of **Os1** was further evidenced by HPLC analysis. As shown in Fig. 1c, the retention time of the new peak is consistent with that of the independently synthesized **Os1** (Supplementary Figs. 6–10, Tables S1–S2. See also SI for the synthesis and characterization of **Os1**). Moreover, after 60 min of irradiation, the transform efficiency of 77.45% from **Os2** to **Os1** demonstrated that **Os1** is the main photoproduct (Fig. 1c), and the transform rate was determined to be 0.46 $\mu mol\,L^{-1}\,min^{-1}$ based on the absorbance changes at 459 nm of **Os2** (Fig. 1b and Supplementary Fig. 11).

**ROS detection in solution.** Due to the dissociation of $O_2$ unit from the metal center of **Os2** under light irradiation, we sought to verify that the $O_2$ unit is released in the form of a superoxide anion ($O_2^{\bullet-}$). We first tried to detect $O_2^{\bullet-}$ by using the non-fluorescent dihydrorhodamine 123 (DHR123) probe[24], which can react with $O_2^{\bullet-}$ and emit strong green fluorescence around 526 nm. As shown in Fig. 2a, when irradiated at 465 nm (13 mW/cm²), the solution of **Os2** showed an increasing emission, suggesting the generation of $O_2^{\bullet-}$. We also simulated the Fenton reaction process[37]. When **Os2** (15 µM) and a small amount of superoxide dismutase (SOD) were added into a PBS solution containing $Fe^{2+}$, the absorbance of methylene blue (MB) probe (MB, 5 µg/mL) at 665 nm decreased with a degradation rate of 70.26% (Supplementary Fig. 12). On the contrary, in control experiments using only **Os2** and **Os2** + SOD, respectively, no changes were observed in the characteristic UV–Vis absorption band of MB. On the basis of these findings and published data, we propose the mechanism in Fig. 2f. Under the action of SOD enzyme, $O_2^{\bullet-}$ disproportionately generates $H_2O_2$ and $O_2$. The OH• is generated from $H_2O_2$ under the action of $Fe^{2+}$, and then reacts with MB, thus decreasing its absorption. These findings confirmed that **Os2** can produce $O_2^{\bullet-}$ under light irradiation.

To gain more insight into the nature of ROS, we made electron spin resonance (ESR) measurements using 5,5-dimethyl-1-pyrro-line-N-oxide (DMPO) and (2,2,6,6-tetramethylpiperidine)oxyl as spin traps. Specifically, DMPO was used to trap $O_2^{\bullet-}$ generated by **Os2** under 465 nm light irradiation[37–39]. After the mixture containing DMPO and **Os2** was exposed to light irradiation, several signal peaks of DMPO-OOH appeared in the 3480–3540 G (Fig. 2b), which explained the generation of $O_2^{\bullet-}$. In order to study whether **Os2** produces $^1O_2$ under light irradiation, we used TEMP to measure $^1O_2$ generation[40,41]. As shown in Fig. 2c, a three-line signal with the intensity of 1:1:1 between 3480 and 3530 G was observed in the ESR spectrum of **Os2** and TEMP mixing solution under light irradiation. A similar signal was observed in the presence of **Os1** (the light product of **Os2**). These ESR results suggest that **Os2** can generate both $O_2^{\bullet-}$ and $^1O_2$ under light irradiation.

In addition, 9,10-anthracenediyl-bis-(methylene) dimalonic acid (ABDA) was further used to detect the $^1O_2$ quantum yields of **Os1** and **Os2** under light irradiation. As shown in Fig. 2d and

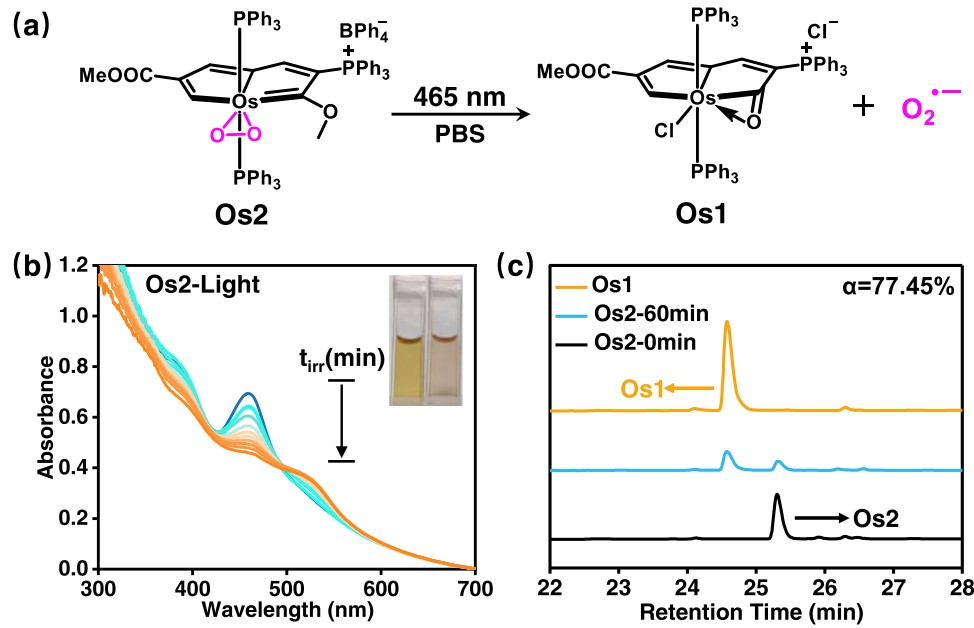

**Fig. 1 Photoactivation of Os2. a** Schematic diagram of light-mediated **Os2** release of $O_2^{\bullet-}$ and **Os1**. **b** The UV–Vis absorption spectra of **Os2** (100 μM) in PBS solution (pH = 7.4, 1% DMSO) under light illumination (465 nm, 13 mW/cm$^2$) at 298 K. **c** HPLC analysis of **Os2** (10 μM) in PBS solution (1% DMSO) in the dark or after irradiating for 60 min. Mobile phase was $CH_3CN$ and water (v/v 1:1).

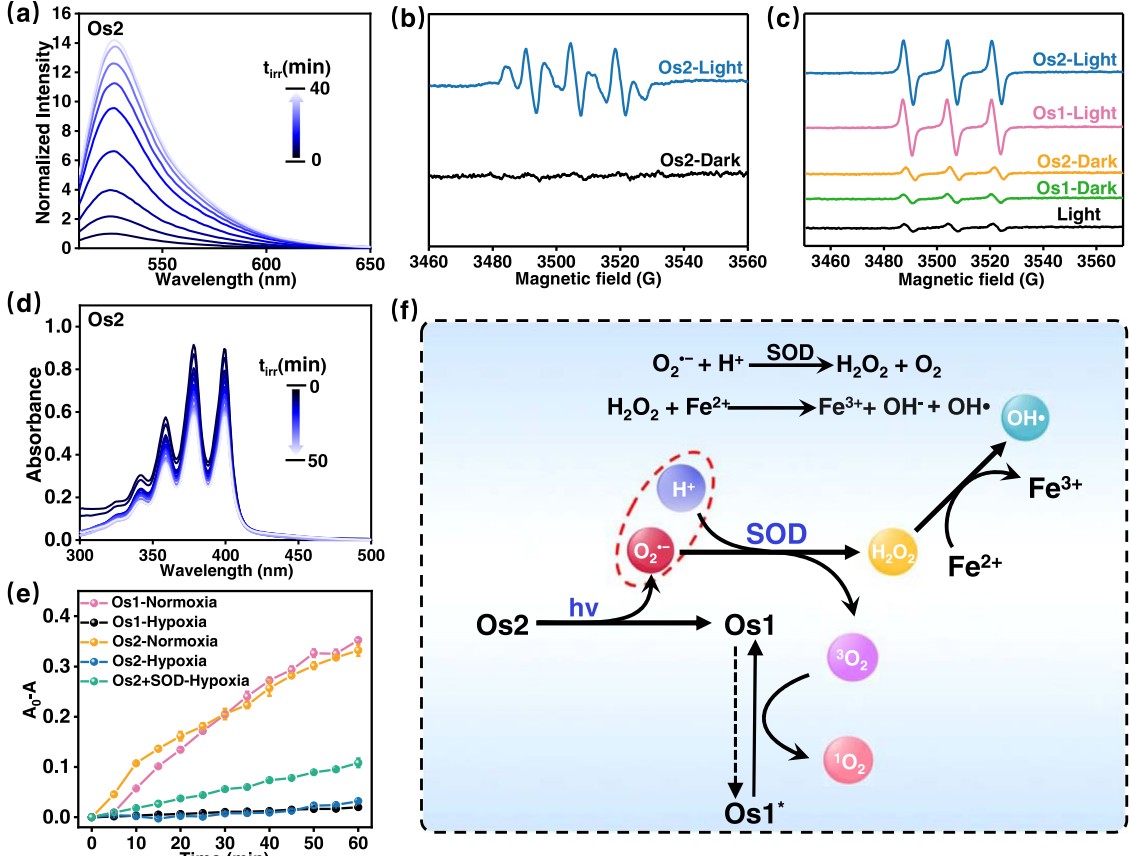

**Fig. 2 Detection of ROS in solution. a** The emission spectra for monitoring $O_2^{\bullet-}$ generation by **Os2** (10 μM) using DHR123 probe (10 μM, $\lambda_{ex}$ = 488 nm) in normoxia. **b** The ESR signal of $O_2^{\bullet-}$ trapped by DMPO (18 mM) after light irradiation in normoxia. **c** The ESR signal of $^1O_2$ trapped by TEMP (8 mM) after light irradiation in normoxia. **d** The UV–Vis absorption spectra of ABDA (100 μM) for monitoring of $^1O_2$ generated by **Os2** (20 μM) under light irradiation in normoxia. **e** The plot of $A_0$-A of ABDA at 378 nm versus irradiation time in the presence of **Os1** or **Os2** in hypoxia or normoxia ($n$ = 3 independent experiments). Error bars represent S.D. from the mean. **f** Schematic diagram of generating $O_2^{\bullet-}$, $^1O_2$ and OH• by light-activating **Os2**. Light irradiation: 465 nm, 13 mW/cm$^2$; Experiment temperature: 298 K; SOD superoxide dismutase.

Supplementary Fig. 13, the characteristic absorption peak of ABDA at about 378 nm decreased gradually as the irradiation time increased. The $^1O_2$ quantum yields of **Os1** and **Os2** were calculated as 0.039 and 0.04 by comparison with $Ru(bpy)_3^{2+}$ (Supplementary Fig. 13 and Table S3)[42].

The above experiments were all conducted under aerobic condition. Next, we studied the production of ROS in the absence of oxygen (hypoxia, deaeration with nitrogen). As shown in Supplementary Fig. 14, the emission spectra showed enhancing green fluorescence in the presence of **Os2** and DHR123 probe. This result proves that **Os2** also produces $O_2^{\bullet-}$ under hypoxia. We further measured whether $^1O_2$ generation under hypoxia. As shown in Supplementary Fig. 15, the absorption of ABDA in the presence of both **Os1** and **Os2** failed to change obviously under hypoxia, confirming that no $^1O_2$ generation occurred under hypoxia. However, the absorption of ABDA was reduced in the presence of **Os2** + SOD. This is due to **Os2** produces $O_2^{\bullet-}$ under hypoxia, and that $O_2^{\bullet-}$ disproportionately generates $H_2O_2$ and $O_2$ under the action of the SOD enzyme, the $O_2$ being available for $^1O_2$ generation by the photoproduct **Os1** (Fig. 2f). However, **Os1** is unable to produce any ROSs under hypoxia (Supplementary Fig. 15).

Our previous studies showed that **Os2** had catalytic activity for alcohol dehydrogenation under relatively harsh conditions (80 °C, $K_2CO_3$, pure oxygen atmosphere), and a concerted double-hydrogen transfer mechanism was proposed based on experimental and theoretical results, but the catalytic reaction hardly proceeds at room temperature[37]. We further verified the stability of **Os2** in the cellular environment before the cellular experiment. The results show that it is highly stable in Dulbecco's Modified Eagle Medium (DMEM) cell culture media or in the PBS solution with different pH values in the dark at 298 K (Supplementary Figs. 16 and 17). Furthermore, treatments of **Os2** with reducing agents (NADH, GSH, and Cys) or cell oxidants (such as $H_2O_2$, peroxidases, cytochrome p450) in the dark did not result in significant changes in the UV–Vis absorption spectra (Supplementary Fig. 18). Supplementary Figs. 19 and 20 further proved that **Os2** does not react with ROS such as $^1O_2$. These results suggest that **Os2** is stable in the cellular environment in the dark.

**Generation of cellular ROS under irradiation**. We studied the photo-induced ROS generation ability of **Os2** in living HeLa cells. Dihydroethidium (DHE), hydroxyphenyl fluorescein (HPF) and singlet oxygen sensor green (SOSG) were used as probes to monitor the production of $O_2^{\bullet-}$, OH• and $^1O_2$, respectively[37,38]. The detection mechanisms of these probes for various free radicals are shown in Fig. 3a. Confocal microscopy showed that when **Os2**-incubated HeLa cells were exposed to light under normoxia (20% $O_2$, 465 nm, 13 mW/cm$^2$, 1 h), the fluorescence signal in DHE-stained cells is significantly enhanced (Fig. 3b, c). This confirms the ability of **Os2** to release $O_2^{\bullet-}$ in cells under light irradiation. Under hypoxia (1% $O_2$), the fluorescence intensity after light irradiation was similar to that obtained under normoxia, indicating that the process of $O_2^{\bullet-}$ generation from **Os2** is independent of $O_2$. This is different from the mechanism of $O_2$-dependent type I photodynamic mechanism.

Because it is known that $O_2^{\bullet-}$ can produce OH• in the presence of intracellular SOD enzyme and $Fe^{2+}$ ions[37], we further studied the generation of OH• in the cells by HPF staining. Confocal imaging showed that **Os2** can produce OH• in HeLa cells under normoxia or hypoxia (Fig. 3b, c). In addition, we also compared the ability of $^1O_2$ generation of **Os2** and the photoproduct **Os1** by using a SOSG probe, and found that under hypoxia, **Os2** has more advantages in producing $^1O_2$ than **Os1**. The result showed that the green fluorescence in the **Os2** treated

cells is strong under both normoxia and hypoxia. However, the fluorescence in the **Os1** treated cells was much weaker under hypoxia than that observed under normoxia (Supplementary Fig. 21). These results are consistent with those in Fig. 2f, showing that **Os2** firstly induces the release of $O_2^{\bullet-}$ and production of **Os1**, accompanied by the formation of $O_2$ (as an oxygen source for **Os1**) and OH• by a Fenton reaction. Finally, $^1O_2$ is generated by the photoproduct **Os1**, but it is difficult to establish whether **Os2** can also produce $^1O_2$ due to its conversion to **Os1** all the time under 465 nm light irradiaton. These data all show that **Os2** may be useful as an effective drug for photoactive treatment of hypoxic tumor cells.

**Phototoxicity in vitro**. The dark- and photo-cytotoxicities of **Os2** and **Os1** against HeLa cells were determined with an MTT method[40,41]. As shown in Fig. 4 and Table 1, **Os2** had low dark-cytotoxicity and high photo-cytotoxicity under both normoxia and hypoxia (Fig. 4a, c). The $(IC_{50})_{dark}$ values, determined with only dark incubation, were 89.2 and >100 μM, respectively (Table 1). The $(IC_{50})_{Light}$ values of **Os2**, determined by photo-irradiation under normoxia or hypoxia were 1.23 and 5.86 μM, respectively. The photo-cytotoxicity under hypoxia can be explained in terms of the release of $O_2^{\bullet-}$ by **Os2** and light irradiation, followed by the production of $O_2$, which may be used by the light product **Os1** to generate $^1O_2$, and OH• with stronger toxicity via the Fenton reaction. In contrast, the light product (**Os1**) shows high dark-cytotoxicity and photo-cytotoxicity under both normoxia and hypoxia (Fig. 4b, d). The $(IC_{50})_{dark}$ values of **Os1** under normoxia and hypoxia are 8.12 and 9.95 μM, and the $(IC_{50})_{Light}$ values under normoxia and hypoxia are 1.31 and 7.51 μM, respectively (Table 1). This result suggests that **Os1** is high photo- and dark- cytotoxic, and can be seen as a chemotherapeutic drug. However, with hypoxia, due to the lack of oxygen, **Os1** shows no obvious photodynamic effect. The PI value of **Os2** (>18) under hypoxia is much higher than that of **Os1** (1.3).

Cisplatin and 5-aminolevulinic acid (5-ALA) were used as controls (Table 1). Very low dark- and photo- toxicities (both $(IC_{50})_{dark}$ and $(IC_{50})_{light}$ > 50 μM) were showed for HeLa cells after 8 h 5-ALA drug exposure and 40 h recovery (the same conditions as **Os2**). Cisplatin has a certain dark toxicity ($(IC_{50})_{dark}$ = 33.22 μM) after 8 hours of administration, and its phototoxicity is not obvious compared with dark toxicity. We further measured that the cellular uptake of **Os2** or cisplatin towards HeLa cancer cells after 2, 4 and 8 h incubation (Supplementary Fig. 22). The results showed that the accumulation of **Os2** in cells was less than that of cisplatin. This cause lower dark toxicity of **Os2** than cisplatin, thus **Os2** is a feasible drug that can reduce toxic and side effects compared with cisplatin in the dark. However, the phototoxicity of **Os2** is much higher than that of cisplatin under irradiation (Table 1).

Staining with calcein acetoxymethyl ester (Calcein-AM) or propidium iodide (PI) were also used to distinguish living cells (green) from dead cells (red). As shown in Supplementary Fig. 23, the dark group of **Os2** in normoxia and hypoxia shows strong green fluorescence (living cells) but no red fluorescence (dead cells), and the control group in the dark or under light irradiation also shows no dead cells. In contrast, the light group of **Os2** shows weak green fluorescence and strong red fluorescence, indicating that **Os2** leads to a large number of dead cells under light irradiation.

**Ferroptosis mechanism**. It has been reported that the photochemical process of ROS generation could cause ferroptosis in cancer cells[6,43,44]. This led us to consider if **Os2** can induce

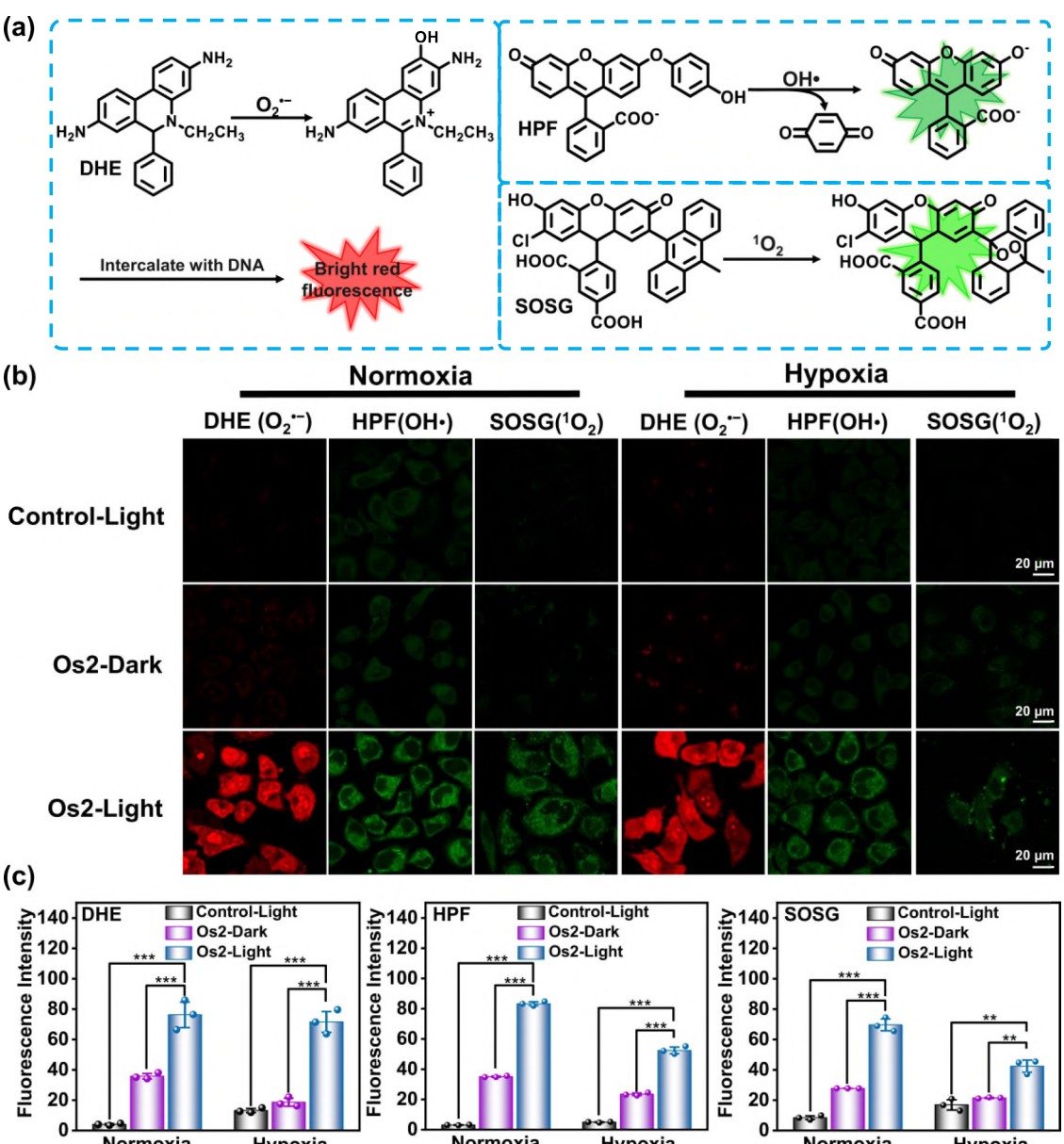

**Fig. 3 Cellular ROS mesurement. a** Schematic diagram of detection of $O_2^{\bullet-}$, OH• and $^1O_2$ with DHE, HPF and SOSG probes, respectively. **b** Confocal microscope images of $O_2^{\bullet-}$, OH• and $^1O_2$ in HeLa cells under normoxia (20% $O_2$) or hypoxia (1% $O_2$) probed by DHE, HPF and SOSG, respectively. **c** The average fluorescence intensities calculated from the images in **b**. All experiments were repeated three times independently with similar results. Error bars represent S.D. from the mean. Statistical significance was calculated with two-tailed Student's t test (*$p < 0.05$, **$p \leq 0.01$ or ***$p \leq 0.001$). HeLa cells incubated with **Os2** (20 μM) for 8 h, and then treated with DHE (10 μM, 30 min), HPF (10 μM, 1 h) or SOSG (2.5 μM, 30 min). Incubation temperature: 310 K; Light irradiation: 465 nm, 13 mW/cm², 1 h; DHE: $\lambda_{ex} = 488$ nm, $\lambda_{em} = 600 \pm 30$ nm; HPF: $\lambda_{ex} = 488$ nm, $\lambda_{em} = 530 \pm 30$ nm; SOSG: $\lambda_{ex} = 488$ nm, $\lambda_{em} = 525 \pm 30$ nm. DHE dihydroethidium; HPF hydroxyphenyl fluorescein; SOSG singlet oxygen sensor green.

ferroptosis. GSH is closely related to ferroptosis[45,46], and we first detected the ability of **Os2** to consume GSH. As shown in Fig. 5a, b, with an increase of light irradiation time, the absorption at 412 nm decreased, indicating that **Os2** when irradiated, could consume GSH. We determined the GSH levels in the cells after treatment with **Os2**, and found that the GSH level in the irradiated group was significantly lower than that in the non-irradiated group (Fig. 5c). Therefore, we concluded that **Os2** can consume cellular GSH under irradiation conditions.

Ferroptosis is a type of iron-dependent cell death caused by excessive lipid peroxidation. The main feature of ferroptosis is that after the inactivation of cell antioxidant capacity,

phospholipids containing polyunsaturated fatty acids are peroxidated on the cell membrane, destroying the cell membrane and leading to ferroptosis[47–49]. The antioxidant glutathione peroxidase 4 (GPX4) specifically catalyzes loss of oxidative activity in the lipid peroxides in a glutathione-dependent manner[50,51], and subsequent inhibition of GPX4 induces ferroptosis. GSH consumption can indirectly inhibit the expression of GPX4. We speculated that **Os2** could further inhibit the expression of GPX4 and we verified this hypothesis by western blot analysis of GPX4. As shown in Fig. 5f–g, **Os2** fails to reduce the expression of GPX4 in the dark. However, upon light irradiation, **Os2** significantly reduces the expression of GPX4, and

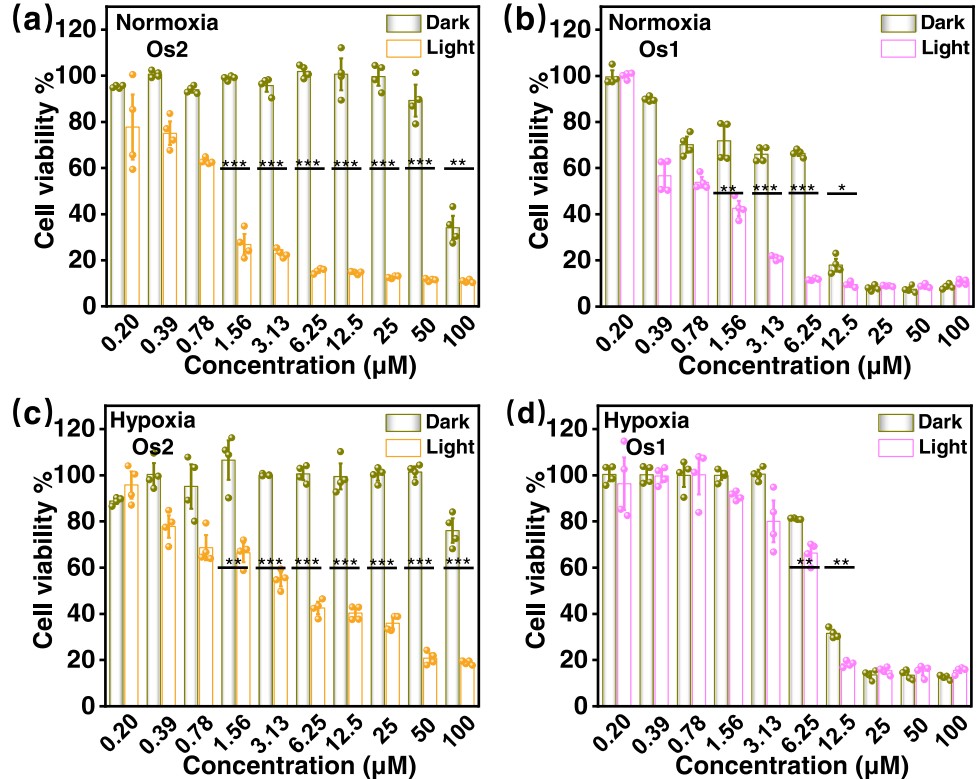

**Fig. 4 Cytotoxicity against HeLa cells.** The viabilities of HeLa cells treated with different concentrations of **Os2** or **Os1** under normoxia (20% O₂, **a**, **b**) or hypoxia (1% O₂, **c**, **d**) in the dark or upon light irradiation. All cell viability data was performed as duplicates of quadruplicate (n = 4 biologically independent samples). Error bars represent S.D. from the mean. Statistical significance was calculated with two-tailed Student's t test (*p < 0.05, **p ≤ 0.01 or ***p ≤ 0.001). Incubation temperature: 310 K. Light irradiation: 465 nm, 13 mW/cm², 1 h.

**Table 1 The dark- and photo-IC₅₀ values of the compounds against HeLa cells under normoxia and hypoxia.**

| Compounds | Condition | Dark [μM] | Light[a] [μM] | PI[b] |
|-----------|-----------|-----------|---------------|-------|
| Os2 | Normoxia (20% O₂) | 89.2 ± 2.6 | 1.23 ± 0.09 | 72.5 |
| Os2 | Hypoxia (1% O₂) | >100 | 5.86 ± 0.19 | >18 |
| Os1 | Normoxia (20% O₂) | 8.12 ± 0.26 | 1.31 ± 0.05 | 6.2 |
| Os1 | Hypoxia (1% O₂) | 9.95 ± 0.14 | 7.51 ± 0.35 | 1.3 |
| 5-ALA | Normoxia (20% O₂) | >100 | 68.62 ± 1.99 | >1.45 |
| Cisplatin | Normoxia (20% O₂) | 33.22 ± 0.77 | 47.75 ± 1.5 | – |

5-ALA is 5-aminolevulinic acid.
[a]Photoirradiation was imposed (465 nm, 13 mW/cm², 1 h) after 8 h of the complexes incubation and then 40 h recovery.
[b]Photo-cytotoxicity index, the ratio of (IC50)dark/(IC50)Light.

the GSH consumption caused by ROS and the inhibition of GPX4 will further lead to the accumulation of lipid peroxides and induce ferroptosis. We used C11-BODIPY as a lipid peroxide probe with which to monitor intracellular accumulation of lipid peroxides (Fig. 5d, e). Confocal microscopy showed that the fluorescence of HeLa cells treated with **Os2** was significantly enhanced after exposure to light, indicating a significant accumulation of lipid peroxides, which could be effectively inhibited by Ferrostatin-1 (Fer-1, a ferroptosis inhibitor). All the above results confirm that **Os2** induces ferroptosis as shown in Fig. 5h.

**NADH photocatalytic oxidation**. As a cofactor, 1,4-dihydro-nicotinamide adenine dinucleotide (NADH) regulates the redox balance of cellular mitochondria, and plays an important role in regulating energy production. If NADH is oxidized to NAD⁺, it can destroy the whole respiratory chain and kill cells[19,52,53]. We studied whether NADH can be oxidized by **Os2** under light

irradiation, which could provide a photocatalytic oxidation pathway to kill cancer cells. The photocatalytic efficiency of **Os2** (20 μM) towards NADH (175 μM) was first determined by UV–Vis absorption spectroscopy. As shown in Fig. 6a, the absorbance at 339 nm decreases and the absorbance at 259 nm increases gradually with increase of the irradiation time. In contrast, the absorption of the non-illuminated **Os2** group exhibits no obvious changes (Supplementary Fig. 24). This indicates that **Os2** can reduce the enzyme activity of NADH under light irradiation. We also calculated the NADH oxidation turnover number (TON) of **Os2** at 339 nm to evaluate its photocatalytic efficiency. The TON value of NADH oxidation by **Os2** under light irradiation is 3.888, which is 30-times higher than that of the non-illuminated group (TON_dark = 0.137) (Fig. 6b). Similarly, **Os1** also shows a similar photocatalytic oxidation effect on NADH (Supplementary Fig. 25).

The photocatalytic oxidation of NADH was also monitored by ¹H NMR in D₂O/CD₃OD (1/3, v/v) at 298 K. As shown in

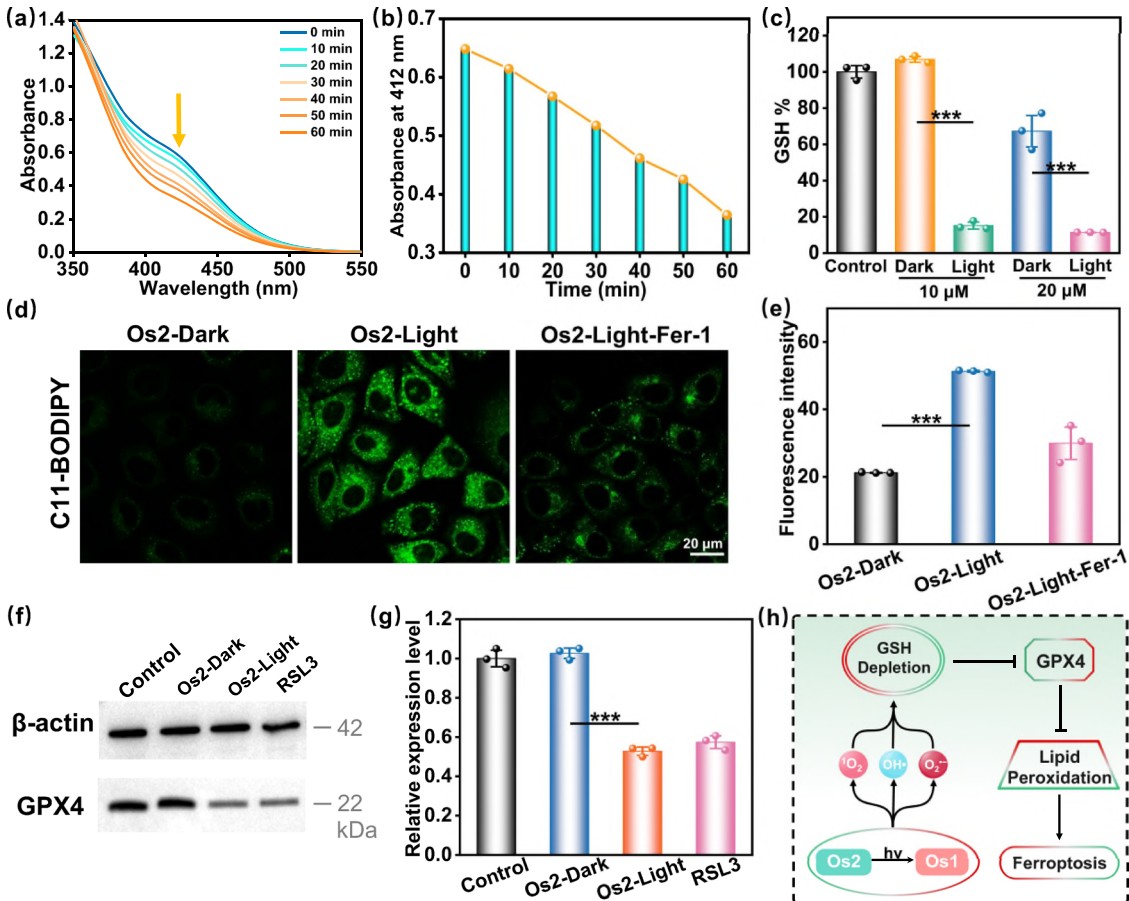

**Fig. 5 Ferroptosis induction by Os2 under light irradiation. a** Irradiation time-dependent GSH (200 μM) depletion by **Os2** (20 μM) upon blue light irradiation at 298 K and **b** the absorption at 412 nm was decreased by increasing the irradiation time. **c** The GSH levels in cells after different treatments. All the experiments were performed as duplicates of triplicates ($n = 3$ biologically independent samples, $p$ values ($p$): 10 μM-0.000025, 20 μM-0.00065). **d** The fluorescence images of lipid peroxides in the treated cells detected by C11-BODIPY probe (30 μM, 310 K, 0.5 h). C11-BODIPY: $\lambda_{ex} = 488$ nm, $\lambda_{em} = 570 \pm 50$ nm. **e** The average fluorescence intensities calculated from the images in **d**. The experiment was repeated three times independently with similar results, $p = 0.000024$. **f** Western blot analysis of GPX4 in HeLa cells after treatment with **Os2** (20 μM, 310 K, 8 h) with or without light treatment. RSL3 is the positive control group. **g** The relative expression levels of GPX4 calculated from **f** All the experiments were performed as duplicates of triplicates ($n = 3$ biologically independent samples, $p = 0.000025$). Error bars represent SD from the mean. Statistical significance was calculated with two-tailed Student's $t$ test (*$p < 0.05$, **$p \leq 0.01$ or ***$p \leq 0.001$). **h** The process of ferroptosis in this photoactive antitumor therapy. Light irradiation: 465 nm, 13 mW/cm$^2$. GSH glutathione, Fer-1 ferrostatin-1, GPX4 glutathione peroxidase 4.

Fig. 6c, In the **Os2** and NADH illuminated group, new peaks from hydrogens on the nicotinamide ring of NAD$^+$ are observed at 6.13, 8.31, 8.55, 8.99, 9.36 and 9.58, but no new NAD$^+$ peak is observed in either the non-illuminated group or the non-**Os2** group. Intuitively, this shows that **Os2** could oxidize NADH under light irradiation, transforming it into NAD$^+$. Subsequently, we measured the NADH photocatalytic oxidation ratio at the cellular level using a NAD/NADH-Glo$^{TM}$ method, which is a bioluminescence method for the detection of NAD$^+$ and NADH. As shown in Fig. 6d, the chemical luminescence intensity of **Os2**-light group proved to be lower than that of other groups. These results show that **Os2** can effectively oxidize NADH at the cellular level by light irradiation, thereby killing cancer cells. As can be seen in Fig. 6e, NADH can not only be converted to NAD$^+$ by interacting with Os1$^*$ (the excited state of **Os1**), but can also be oxidized to NAD$^+$ by $O_2^{\bullet-}$ released from **Os2**. The down-regulation of NADH indirectly aids the reduction of oxidized glutathione (GSSG) to GSH by glutathione reductase[54], resulting in the accumulation of lipid peroxides, and ultimately achieving the synergistic induction of ferroptosis.

**Photoactive antitumor therapy in vivo**. We studied the feasibility of photoactive therapy in vivo by Os2. Since Os2 is a small molecule and there is no specific targeted group, it can be administered intratumorally (Fig. 7a). As shown in Fig. 7b, compared with the other three groups, the tumor growth in mice treated with Os2-light group was inhibited. The tumor size of the Os2-light group was the smallest in the four groups (Fig. 7d), and the average tumor weight of the Os2-light group was significantly lower than that of the other groups (Fig. 7c). The tumor tissues after final treatment were collected for histological assessment. The hematoxylin and eosin (H&E) staining showed obvious destruction of tumor tissues in the Os2-light group, while tumor tissues in the other three groups were not affected (Fig. 7e).

In order to evaluate the biological safety of Os2, we first analyzed the H&E staining slices of the main organs of healthy mice i.v. injected with three times of the therapeutic dose (2.69 mg kg$^{-1}$). The results showed no obvious tissue damage in these slices (Supplementary Fig. 26). We also used zebrafish to test the biological safety of Os2 (Supplementary Fig. 27) with the green fluorescent protein (GFP) commonly used as a biomarker to visualize the physiological processes. After 5 days of incubation

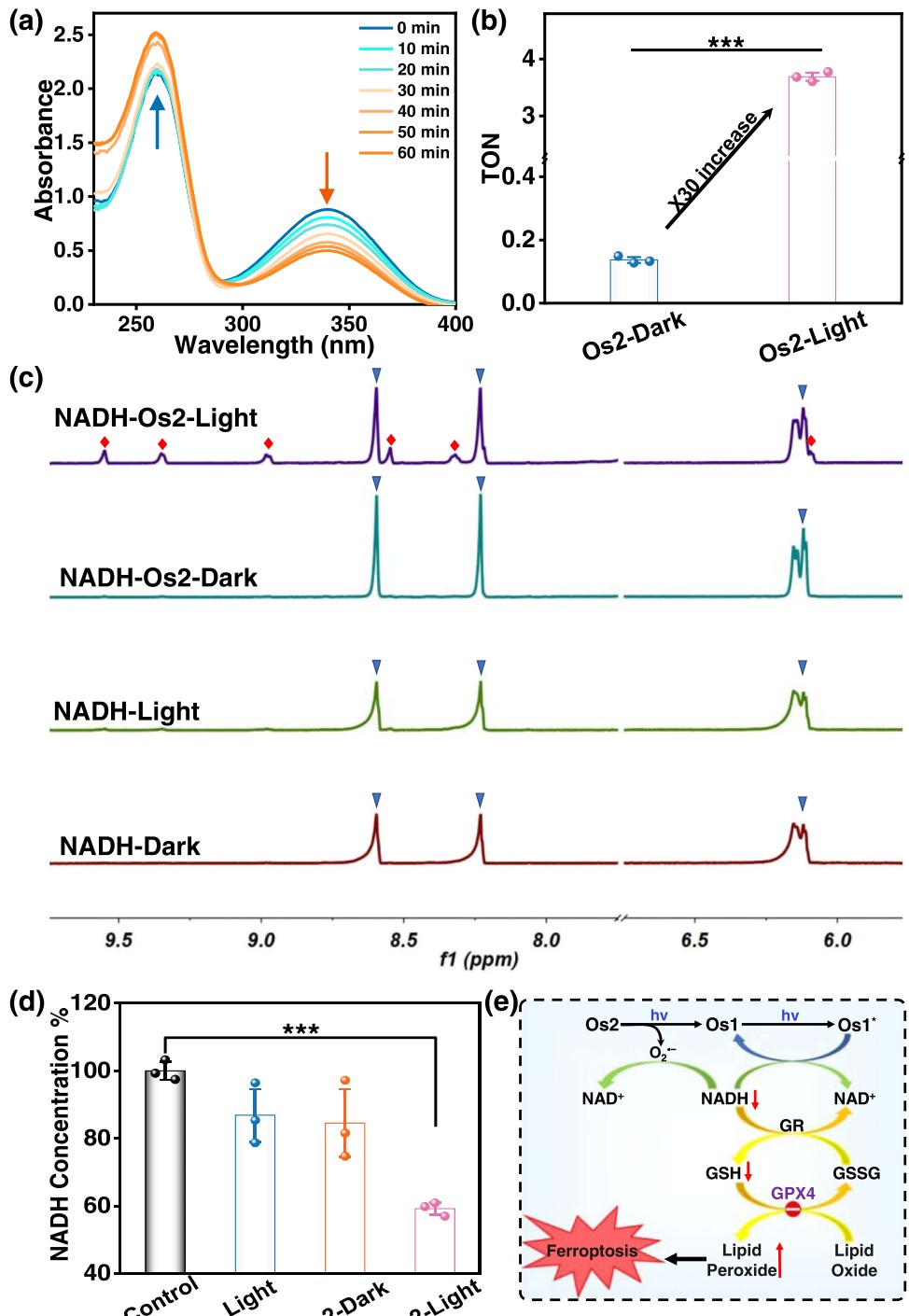

**Fig. 6 Photocatalytic oxidation of NADH by Os2 under light irradiation. a** Reaction of **Os2** (20 μM) and NADH (175 μM) in PBS solution under blue light irradiation monitored by UV–Vis absorption spectra at 298 K. **b** TON of **Os2** under dark or irradiation conditions ($n = 3$ independent experiments, $p = 0.0000016$). **c** Photocatalytic oxidation of NADH (3.5 mM) by **Os2** (0.25 mM) under dark or irradiation conditions monitored by [1]H NMR spectroscopy at 298 K. Peaks associated with blue triangles represent NADH, those with red squares represent NAD[+]. **d** NADH concentrations in the treated HeLa cells. Os2: 20 μM, 310 K, 8 h. All the experiments were performed as duplicates of triplicates ($n = 3$ biologically independent samples, $p = 0.000039$). Error bars represent S.D. from the mean. Statistical significance was calculated with two-tailed Student's $t$ test (*$p < 0.05$, **$p \leq 0.01$ or ***$p \leq 0.001$). **e** Schematic diagram of the photocatalytic oxidation of NADH by **Os2** and followed by induced ferroptosis. Light irradiation: 465 nm, 13 mW/cm². TON turnover number, NADH 1,4-dihydro-nicotinamide adenine dinucleotide, GSH glutathione; GSSG oxidized glutathione, GPX4 glutathione peroxidase 4, GR glutathione reductase.

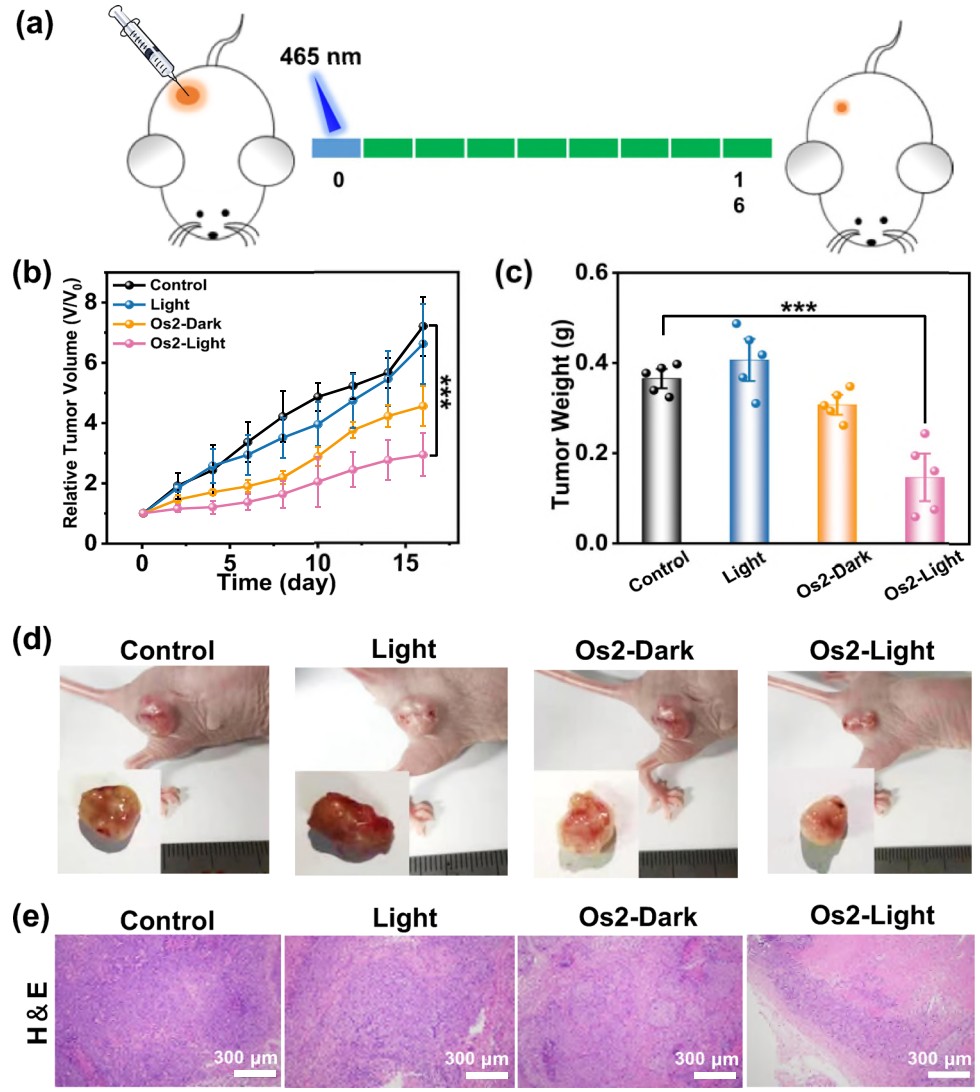

**Fig. 7 In vivo photoactive chemotherapy. a** Schematic of the in vivo (HeLa tumor-bearing Balb/c mice) therapeutic protocol. Mice were irradiated by light (465 nm, 13 mW/cm²) for 60 min after i.t. injection with 25 μL of PBS containing 500 μM Os2. **b** Tumor growth curves after treatment. Error bars were standard errors (±SD) based on five mice in each group. Statistical significance was calculated with two-tailed Student's $t$ test, $p = 0.0004$ (*$p < 0.05$, **$p \leq 0.01$ or ***$p \leq 0.001$). **c** Tumor weights of mice at day 16 after various treatments. Error bars were standard errors (±SD) based on five mice in each group. Statistical significance was calculated with two-tailed Student's $t$ test, $p = 0.00042$ (*$p < 0.05$, **$p \leq 0.01$ or ***$p \leq 0.001$). **d** The digital photos of representative mice after various treatments. **e** H&E staining images of HeLa tumor tissues in mice after various treatments. The experiment was repeated three times independently with similar results.

with Os2, the blood vessels of the zebrafish were apparently not damaged, showing good biocompatibility in vivo.

## Discussion

In summary, we have demonstrated that an osmium-peroxo complex (**Os2**) can release $O_2^{\bullet-}$ under light irradiation in the absence of $O_2$, and at the same time is transformed into another active osmium complex (**Os1**), which exhibits both chemotherapeutic and photodynamic properties, thus maintaining good phototoxicity in hypoxic tumors. The osmium-peroxo complex **Os2** can induce ferroptosis, which is characterized by GSH degradation, GPX4 down-regulation and lipid peroxide accumulation. In addition, under light irradiation, the same osmium-peroxo complex oxidizes NADH into $NAD^+$, further helping induction of ferroptosis. At the in vivo level, the osmium-peroxo complex achieves highly effective photoactive therapy of solid

hypoxic tumors. This study reports an interesting example of a metal-peroxo complex for $O_2$-independent photoactive therapy and provides a promising strategy for combating hypoxic tumors.

## Methods

### ROS detection in solution

$O_2^{\bullet-}$ *detection*. Dihydrorhodamine 123 (DHR123) was used as the superoxide anion radical indicator, which can be converted to Rhodamine 123 in the presence of $O_2^{\bullet-}$. **Os2** (10 μM) and DHR123 (10 μM) were mixed. Then the cuvette was exposed to 465 nm monochromatic light for different time, and the fluorescence spectra were observed immediately after each irradiation. The $O_2^{\bullet-}$ generation was studied in normoxia (20% $O_2$) and hypoxia (deaeration with nitrogen, <1% $O_2$), respectively.

$^1O_2$ *detection*. 20 μM **Os1** or **Os2** and 100 μM ABDA mixing solution was measured using UV–Vis spectrophotometer after different light (465 nm, 13 mW/cm²) irradiation durations. The absorbance changes of ABDA at 378 nm were recorded to quantify the quantum yields (Φ) of $^1O_2$. The $^1O_2$ generation was studied in normoxia (20% $O_2$) and hypoxia (deaeration with nitrogen, <1% $O_2$), respetively.

$OH\bullet$ detection. 15 μM the osmium complex and 5 μg mL$^{-1}$ MB mixing solution (pH = 4.5 containing $Fe^{2+}$ (0.2 mM)) was measured using UV–vis spectrophotometer after different light (465 nm, 13 mW/cm$^2$) irradiation durations.

## Intracellular ROS measurement

*Intracellular $O_2^{\bullet-}$ measurement.* HeLa cells were incubated with 20 μM **Os2** for 8 h followed by incubation with 10 μM DHE for another 30 min under hypoxia (1% $O_2$) or normoxia (20% $O_2$). After that, cells were irradiated with 465 nm blue light for 1 h at a power density of 13 mW/cm$^2$. The red fluorescence was immediately observed using CLSM with the excitation wavelength of 488 nm, and emission collection wavelength from 570 to 630 nm.

*Intracellular $OH\bullet$ measurement.* HeLa were incubated with 20 μM **Os2** for 8 h followed by incubation with 10 μM HPF for 1 h under hypoxia (1 % $O_2$) or normoxia (20% $O_2$). After that, cells were washed with PBS and then irradiated with 465 nm blue light for 1 h at a power density of 13 mW/cm$^2$. The green fluorescence was immediately observed using CLSM with the excitation wavelength of 488 nm, and emission collection wavelength from 500 to 560 nm.

*Intracellular $^1O_2$ measurement.* HeLa cells were incubated with 20 μM **Os2** or **Os1** for 8 h followed by incubation with 2.5 μM SOSG for 30 min under hypoxia (1% $O_2$) or normoxia (20% $O_2$). After that, cells were washed with PBS and then irradiated with 465 nm blue light for 1 h at a power density of 13 mW/cm$^2$. The green fluorescence was immediately observed using CLSM with the excitation wavelength of 488 nm, and emission collection wavelength from 495 to 555 nm.

**Reporting summary.** Further information on research design is available in the Nature Research Reporting Summary linked to this article.

## Data availability

The authors declare that all data needed to evaluate the conclusion of this work are presented in the paper, the supplementary information or source data file. The source data have been deposited in the Figshare database under accession code https://doi.org/10.6084/m9.figshare.19333802 [https://figshare.com/articles/figure/Daet_of_An_osmium-peroxo_complex_for_photoactive_therapy_of_hypoxic_tumors_/19333802]. The crystal structure of Os1 was deposited in the Cambridge Crystallographic Data Centre (CCDC 1913382), and the data can be obtained free of charge via www.ccdc.cam.ac.uk/structures.

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

## Acknowledgements

We appreciate the financial support of the National Natural Science Foundation of China (22077085 for P.Z., 22007104 and 21931002 for H.X.), and the Science and Technology Foundation of Shenzhen (JCYJ20210324095200002 and JCYJ20190808153209537 for P.Z., JCYJ20200109140812302 for H.X.). We appreciate the Instrumental Analysis Center of Shenzhen University for the assistance with confocal microscopy analysis.

## Author contributions

N.L., J.G., H.X., and P.Z. designed the study. Z.D. synthesized and characterized the complexes. N.L. and C.L. performed the experiments in vitro and in vivo. N.L., Z.D., J.G., H.X., and P.Z. analyzed the data and wrote the paper. All authors contributed to the general discussion.

## Competing interests

The authors declare no competing interests.
