## [Peer Review File · Nature Communications]

REVIEWER COMMENTS

Reviewer #1 (Remarks to the Author):

Zhang and co-workers report in this paper on "An osmium-peroxo complex for 1 photoactive therapy of hypoxic tumors". Both in vitro and in vivo effects of the Os complex are provided. My comments to the manuscript:

1. Os²⁺ was reported as catalysts for alcohol aerobic oxidation reaction via a non-radical mechanism (iScience 2020, 23, 101379). But the author didn't discuss why this catalytic action didn't work in this paper.
2. The authors confirmed that Os²⁺ is activated by light (Figure 2). Other present in cells oxidants (ROS, peroxidases, cytochrome p450 etc) or pH can also potentially lead the same reaction?. Data on the stability of Os²⁺ in the these conditions are not provided. Therefore, the mechanism of activation is not yet established by the authors.
3. Unfortunately, the complexes described in the paper were not characterized sufficiently well. In particular, no elemental (C, H, N) analysis was provided.
3. In Figure 1b and c, the transform efficiency and rate from Os²⁺ to Os¹⁺ didn't measure.
4. As an inorganic medicine, measuring the total cellular uptake amount is necessary. It can use to evaluate the cytotoxicity with the other chemotherapy drug.
5. As discussed in line 167-170, the author speculates that it is Os¹⁺ generated singlet oxygen rather than Os²⁺. This statement needs more hints. Whether the peroxo structure can quench the singlet oxygen generation?
6. In NADH photocatalytic oxidation experiment, the TON of Os²⁺ under irradiation conditions is 3.808, which is low.
7. In the in vivo experiment, the penetration of 465 nm light may significantly limit the therapeutic effect and the further application of Os complexes. The author didn't use the clinic drug as a control. Why doesn't the author use the mice's body weight to evaluate the biosafety?
8. Throughout the paper captions to Figures should be improved by including experimental details: reagent concentrations, incubation times, temperature, at which experiments were conducted etc.
9. Authors compare average values with each other without mentioning any standard deviations, statistical parameters). This is valid also for the SI: e.g. Table S3 contains Singlet oxygen quantum yields data, which are just values without any standard deviations.

Reviewer #2 (Remarks to the Author):

The manuscript „An osmium-peroxo complex for photoactive therapy of hypoxic tumors” by Nong Lu et al. The manuscript focuses on the use of the osmium complex compound in photoactive therapy. Although this coordination compound was already known along with the procedure of its synthesis, the authors proposed its new use in anti-cancer therapy. The enormous amount of work put into understanding the mechanism of action of the osmium complex compound in anti-cancer therapy deserves praise. The novelty of this work is definitely the confirmation of the anti-cancer effect in photoactive therapy by the metal-peroxo complex for O₂-independent conditions. The obtained osmium complex compounds were well characterized, and their structures and compositions were confirmed by several methods.

The manuscript is very interesting, but while reading it, I came up with a few remarks:

- 1) in the introduction part there is no reference to information on complex metal compounds which are not peroxo but used in photoactive therapy under oxygen deficiency;
- 2) Figure 1 , panel a) instead of „LIGHT”, please specify whether it is Uv, UV-Vis or a specific wavelength;
- 3) some of the complex compounds can generate superoxide anion radical and at the same time, they can play role as an antioxidant. Has this been checked for Os1 and Os2?
- 4) Figure 3, panel a) it is not clear since the arrow pointing to the interpellation with DNA runs, please correct it to make it unambiguous;
- 5) „Phototoxicity in vitro” Section - there is no comparison to the cytotoxicity of other preparations currently used on the market in photoactive therapy. Please complete it.

Summing up my considerations, in my opinion this report is very good despite a few shortcomings that have been mentioned above. After minor revision, I recommend this manuscript for publication in Nature Communications.

Reviewer #1 (Remarks to the Author):

Zhang and co-workers report in this paper on "An osmium-peroxo complex for 1 photoactive therapy of hypoxic tumors". Both in vitro and in vivo effects of the Os complex are provided. My comments to the manuscript:

Reply to the reviewer: We thank the reviewer for the enthusiastic comments. Our responses to these issues are as follows.

1).Os2 was reported as catalysts for alcohol aerobic oxidation reaction via a non-radical mechanism (iScience 2020, 23, 101379). But the author didn't discuss why this catalytic action didn't work in this paper.

Reply to the reviewer: Following the suggestion, we added the relevant discussions in the last paragraph on page 6 of the revised manuscript: “Our previous studies showed that **Os2** had catalytic activity for ethanol dehydrogenation under relatively harsh conditions (80 °C, K₂CO₃, pure oxygen atmosphere), and a concerted double-hydrogen transfer mechanism was proposed based on experimental and theoretical results, but the catalytic reaction hardly proceeds at room temperature.”

In addition, control experiments in this work showed that **Os2** is highly stable in DMEM cell culture media and PBS solutions with different pH values in the dark at room temperature (**Supplementary Figs. 16, 17**). Furthermore, treatments of **Os2** with reducing agents (NADH, GSH and Cys) or cell oxidants (such as H₂O₂, peroxidases, cytochrome p450) in the dark did not result in significant changes in the UV-Vis absorption spectra (**Supplementary Fig. 18**). These results rule out the catalytic action of **Os2** in the biological conditions in the dark.

2).The authors confirmed that Os2 is activated by light (Figure 2). Other present in cells oxidants (ROS, peroxidases, cytochrome p450 etc) or pH can also potentially lead the same reaction?. Data on the stability of Os2 in the these conditions are not provided. Therefore, the mechanism of activation is not yet established by the authors.

Reply to the reviewer: We thank the reviewer for this suggestion. We investigated the stabilities of

Os2 to the mentioned cells oxidants and pH, and the results have been added in the revised manuscript and **Supplementary Figs. 18b** and **17**. The results show that **Os2** is highly stable in the presence of H₂O₂, peroxidases and cytochrome p450, and is also persistent in the PBS solutions with pH = 4-9.

Furthermore, to determine whether ROS can trigger the conversion of **Os2**, we carried out a series of experiments (**Supplementary Figs. 19** and **20**). **Os2** under 633 nm red light irradiation cannot produce ¹O₂ (**Supplementary Figs. 19d**). Ce6 under 633 nm red light irradiation can produce ¹O₂, which results in the degradations of the absorption bands of ABDA (**Supplementary Figs. 19a** and **19f**). However, the additions of **Os2** to Ce6 solution have negligible effects on the degradation rates of the absorption bands of ABDA (**Supplementary Figs. 19b** and **19i**), and **Os2** remain intact (**Supplementary Figs. 20a** and **20b**), indicating that **Os2** cannot react with ¹O₂.

Supplementary Figure 17. The UV-Vis absorption spectra of **Os2** (100 μM) in the PBS solution with different pH values at 298 K.

Supplementary Figure 18. (b) The UV-Vis absorption spectra of Os2 (100 μM) in the presence of cells oxidants (50 μM H_2O_2 , 500 $\mu\text{g}/\text{mL}$ Catalase, Liver Microsomes with 50 nM Cytochrome P450) in the dark at 298 K.

Supplementary Figure 19. (a-e) The UV-Vis absorption spectra of ABDA in the presence of (a) Ce6, (b) Ce6+Os2, (c) Ce6 +Os1, (d) Os2 or (e) Os1 for monitoring of $^1\text{O}_2$ generation under 633 nm light

irradiation. (f-h) The UV-Vis absorption spectra of ABDA in the presence of (f) Ce6, (g) **Os2** or (h) **Os1** for monitoring of $^1\text{O}_2$ generation in the dark. (i) Rate constant for $^1\text{O}_2$ generation according to the absorbance of ABDA at 378 nm in (a-c). **Ce6**: 5 μM ; **Os2** or **Os1**:10 μM ; Light: 633 nm, 6.5 mW/cm^2 .

Ce6 can produce $^1\text{O}_2$ under 633 nm red light irradiation. However, **Os2** cannot generate $^1\text{O}_2$ under 633 nm red light irradiation. When we added both **Os2** and Ce6 in the ABDA solution and then the mixture was irradiated under 633 nm light. The results show that the absorption of ABDA decreases in Ce6 + 633 nm + **Os2** group, which is basically the same as that in Ce6 + 633 nm group, indicating that the addition of **Os2** do not react with (or quench) $^1\text{O}_2$. A similar result occurs with **Os1**.

Supplementary Figure 20. (a, c) The UV-Vis absorption spectra of **Os2** or **Os1** (100 μM) in PBS solution (pH 7.4) containing Ce6 (5 μM) in the dark at 298 K. (b, d) The UV-Vis absorption spectra of **Os2** or **Os1** (100 μM) in PBS solution (pH 7.4) containing Ce6 (5 μM) under light irradiation at 298 K. Light: 633 nm, 6.5 mW/cm^2 .

Ce6 was added to the **Os2** solution and irradiated with 633 nm light, and then the UV-Vis

absorption curves at different irradiation times were measured. The results show that there are no obvious changes in the absorption spectra of **Os2**, meaning that the $^1\text{O}_2$ produced by Ce6 does not react with **Os2**. A similar result occurs with **Os1**.

3). Unfortunately, the complexes described in the paper were not characterized sufficiently well. In particular, no elemental (C, H, N) analysis was provided.

Reply to the reviewer: Following the suggestion, we carried out elemental analysis, and the results have been added to the experimental section of the revised supporting information and highlighted in yellow.

4). In Figure 1b and c, the transform efficiency and rate from **Os2** to **Os1** didn't measure.

Reply to the reviewer: We thank the reviewer for this suggestion. We calculated the transform rate by a linear fitting of the absorption at 459 nm depending on the irradiation time in **Fig. 1b**, the obtained transform rate was $0.46 \mu\text{mol}\cdot\text{L}^{-1}\cdot\text{min}^{-1}$ (**Supplementary Fig. 11**). We also calculated the peak area of **Os2** in the dark (0 min) and after irradiation (60 min) in **Fig. 1c**, the transform efficiency of **Os2** was 77.45 %. These results show that **Os1** is one of the main photoproducts of **Os2**. The results have been added to the end of the photoactivation property section in the revised manuscript and highlighted in yellow.

Supplementary Figure 11. The transform rate of the band at 459 nm for **Os2** under light irradiation in **Fig. 1b**.

5). As an inorganic medicine, measuring the total cellular uptake amount is necessary. It can use to evaluate the cytotoxicity with the other chemotherapy drug.

Reply to the reviewer: We sincerely thank the reviewer for this suggestion. As suggested, we measured the cellular uptake of **Os2** and cisplatin toward HeLa cancer cells after 2, 4 and 8 hours incubation, respectively (**Supplementary Figure 22**), and the results have been added in the second paragraph of Phototoxicity in vitro section in the revised manuscript. As shown in **Supplementary Figure 22**, the accumulation of **Os2** in cells was less than that of cisplatin. This cause lower dark-toxicity of **Os2** than cisplatin (**Table 1**), thus **Os2** is a feasible drug that can reduce toxic and side effects compared with cisplatin in the dark. However, the phototoxicity of **Os2** is much higher than that of cisplatin under irradiation (**Table 1**).

Supplementary Figure 22. Cellular uptake of **Os2** (10 μ M) or cisplatin (10 μ M) in HeLa cells measured by ICP-MS. Error bars represent S.D. from the mean. Statistical significance was calculated with two-tailed Student's t test (* $p < 0.05$, ** $P \leq 0.01$ or *** $P \leq 0.001$).

Table 1. The dark- and photo- IC_{50} values of the compounds against HeLa cells under normoxia and hypoxia.

Compounds	Condition	Dark [μ M]	Light ^[a] [μ M]	PI ^[b]
Os2	Normoxia (20% O ₂)	89.2 \pm 2.6	1.23 \pm 0.09	72.5
Os2	Hypoxia (1% O ₂)	>100	5.86 \pm 0.19	>18
Os1	Normoxia (20% O ₂)	8.12 \pm 0.26	1.31 \pm 0.05	6.2
Os1	Hypoxia (1% O ₂)	9.95 \pm 0.14	7.51 \pm 0.35	1.3

5-ALA	Normoxia (20% O ₂)	>100	68.62±1.99	>1.45
Cisplatin	Normoxia (20% O ₂)	33.22±0.77	47.75±1.5	-

^[a]photoirradiation was imposed (465 nm, 13 mW/cm², 1 h) after 8 h of the complexes incubation and then 40 h recovery; ^[b]photocytotoxicity index, the ratio of (IC₅₀)_{dark}/(IC₅₀)_{Light}. 5-ALA is 5-aminolevulinic acid.

6). As discussed in line 167-170, the author speculates that it is Os1 generated singlet oxygen rather than Os2. This statement needs more hints. Whether the peroxo structure can quench the singlet oxygen generation?

Reply to the reviewer: We are very sorry for the unclear expression in line 167-170. Now, we modify these sentences as follows: “Finally, ¹O₂ is generated by the photoproduct Os1, but it is difficult to establish whether Os2 can also produce ¹O₂ due to its conversion to Os1 all the time under 465 nm light irradiation.”

In addition, as mentioned in the reply to the second question, the peroxo structure (**Os2**) cannot quench the singlet oxygen generation. We carried out a series of experiments (**Supplementary Fig. 19**). **Os2** under 633 nm red light irradiation cannot produce ¹O₂ (**Supplementary Figs. 19d**). Ce6 under 633 nm red light irradiation can produce ¹O₂, which results in the degradations of the absorption bands of ABDA (**Supplementary Fig. 19a**). However, the additions of **Os2** to Ce6 solution have negligible effects on the degradation rates of the absorption bands of ABDA (**Supplementary Figs. 19b**), indicating that **Os2** cannot quench the singlet oxygen generation.

Supplementary Figure 19. (a-e) The UV-Vis absorption spectra of ABDA in the presence of (a) Ce6, (b) Ce6+Os2, (c) Ce6 +Os1, (d) Os2 or (e) Os1 for monitoring of $^1\text{O}_2$ generation under 633 nm light irradiation. (f-h) The UV-Vis absorption spectra of ABDA in the presence of (f) Ce6, (g) Os2 or (h) Os1 for monitoring of $^1\text{O}_2$ generation in the dark. (i) Rate constant for $^1\text{O}_2$ generation according to the absorbance of ABDA at 378 nm in (a-c). **Ce6:** 5 μM ; **Os2** or **Os1:**10 μM ; Light: 633 nm, 6.5 mW/cm^2 .

7). In NADH photocatalytic oxidation experiment, the TON of Os2 under irradiation conditions is 3.808, which is low.

Reply to the reviewer: After our repeated experiments and calculations, the TON value of Os2 is indeed around 3.888 under 465 nm irradiation, but it is much higher (by 30 times) than that in the dark condition ($\text{TON}_{\text{dark}} = 0.137$). Therefore, we believe that Os2 photocatalytically oxidizes NADH

under irradiation.

8). *In the in vivo experiment, the penetration of 465 nm light may significantly limit the therapeutic effect and the further application of Os complexes. The author didn't use the clinic drug as a control. Why doesn't the author use the mice's body weight to evaluate the biosafety?*

Reply to the reviewer: We thank the reviewer for this suggestion. Firstly, the 465 nm light might limit the therapeutic effect and further application of Os2 *in vivo*, so our future work will focus on optimizing the geometric structure to obtain Os complexes that can be activated by deep penetrating near infrared light or ultrasonic. Secondly, in the *in vitro* experiments, we used the clinic photosensitizer 5-ALA (5-Amin-levulinic acid) and clinical chemotherapy drug cisplatin as positive controls. The results showed that 5-ALA has low dark- and photo- toxicities (both $(IC_{50})_{\text{dark}}$ and $(IC_{50})_{\text{light}} > 50 \mu\text{M}$) towards HeLa cells after 8 hours drug exposure and 40 hours recovery (the same conditions as Os2). Cisplatin has a certain dark toxicity ($(IC_{50})_{\text{dark}} = 33.22 \mu\text{M}$) after 8 hours of administration, and its photo-toxicity is not obvious compared with dark-toxicity (**Table 1**). In the *in vivo* experiments, to the best of our knowledge, there is no clinical photosensitizer with definite therapy for cervical cancer under blue light irradiation. Thus, we have no positive control *in vivo*. Lastly, during the monitoring process of biosafety, we did not find that mice were significantly thin. As a result, we paid more attention to the damage of organs in mice to evaluate biosafety.

9). *Throughout the paper captions to Figures should be improved by including experimental details: reagent concentrations, incubation times, temperature, at which experiments were conducted etc.*

Reply to the reviewer: Following the suggestion, the experiment details have been added to the captions of Figures and highlighted in yellow.

10). *Authors compare average values with each other without mentioning any standard deviations, statistical parameters). This is valid also for the SI: e.g. Table S3 contains Singlet oxygen quantum yields data, which are just values without any standard deviations.*

Reply to the reviewer: We apologize for the oversights in the manuscript. As suggested, we have added the standard deviations and statistical parameters in the revised manuscript and SI.

Reviewer #2 (Remarks to the Author):

The manuscript "An osmium-peroxo complex for photoactive therapy of hypoxic tumors" by Nong Lu et al. The manuscript focuses on the use of the osmium complex compound in photoactive therapy. Although this coordination compound was already known along with the procedure of its synthesis, the authors proposed its new use in anti-cancer therapy. The enormous amount of work put into understanding the mechanism of action of the osmium complex compound in anti-cancer therapy deserves praise. The novelty of this work is definitely the confirmation of the anti-cancer effect in photoactive therapy by the metal-peroxo complex for O₂-independent conditions. The obtained osmium complex compounds were well characterized, and their structures and compositions were confirmed by several methods.

The manuscript is very interesting, but while reading it, I came up with a few remarks:

Reply to the reviewer: We thank the reviewer for the positive comments. Our responses to these issues are as follows.

1) *in the introduction part there is no reference to information on complex metal compounds which are not peroxo but used in photoactive therapy under oxygen deficiency;*

Reply to the reviewer: Following the reviewer's suggestion, we added a brief introduction of other metal complexes used in photoactive therapy under oxygen deficiency in the revised manuscript (the second paragraph in the introduction section): "It has been reported that some metal complexes show excellent photodynamic therapy effect toward hypoxic tumors.²⁰⁻²⁶ Most of them do not depend on oxygen through traditional type II mechanism for PDT. For example, Sadler et al. studied that an iridium (III) photoredox catalysis can provide an oxygen-independent mechanism of action to combat hypoxic tumors. The iridium (III) complex photocatalytically oxidized 1,4-dihydro-nicotinamide adenine dinucleotide (NADH), an important coenzyme in living cells, to

generate NAD• radicals with a high turnover frequency, and synergistically photoreduced cytochrome C under hypoxia.²⁰ Chao et al. designed an iridium (III) complex that upon irradiation, produces free carbon radicals under hypoxia.²⁶”

2) *Figure 1, panel a) instead of „LIGHT”, please specify whether it is Uv, UV-Vis or a specific wavelength;*

Reply to the reviewer: Thank the reviewer for nicely pointing out the ambiguous description. We have corrected “LIGHT” to “465 nm” in the revised manuscript.

3) *some of the complex compounds can generate superoxide anion radical and at the same time, they can play role as an antioxidant. Has this been checked for Os1 and Os2?*

Reply to the reviewer: Reviewer 1 also showed comments on the reactivities of the Os2 toward cell oxidants (H₂O₂, ROS, peroxidases, cytochrome p450 etc) (**Supplementary Fig. 18b**). The results indicated that the osmium complexes cannot act as antioxidants under our biological conditions. We have added a discussion on the page 6 of the revised manuscript.

To determine whether ROS can react with **Os2** or **Os1**, we carried out a series of experiments (**Supplementary Figs. 19 and 20**). **Os2** under 633 nm red light irradiation cannot produce ¹O₂ (**Supplementary Figs. 19d**). Ce6 under 633 nm red light irradiation can produce ¹O₂, which results in the degradations of the absorption bands of ABDA (**Supplementary Figs. 19a and 19f**). However, the additions of **Os2** to Ce6 solution have negligible effects on the degradation rates of the absorption bands of ABDA (**Supplementary Figs. 19b and 19i**), and **Os2** remain intact (**Supplementary Figs. 20a and 20b**), indicating that **Os2** cannot react with ¹O₂.

Supplementary Figure 18. (b) The UV-Vis absorption spectra of Os2 (100 μM) in the presence of cells oxidants (50 μM H_2O_2 , 500 $\mu\text{g}/\text{mL}$ Catalase, Liver Microsomes with 50 nM Cytochrome P450) in the dark at 298 K.

Supplementary Figure 19. (a-e) The UV-Vis absorption spectra of ABDA in the presence of (a) Ce6, (b) Ce6+Os2, (c) Ce6 +Os1, (d) Os2 or (e) Os1 for monitoring of $^1\text{O}_2$ generation under 633 nm light

irradiation. (f-h) The UV-Vis absorption spectra of ABDA in the presence of (f) Ce6, (g) **Os2** or (h) **Os1** for monitoring of $^1\text{O}_2$ generation in the dark. (i) Rate constant for $^1\text{O}_2$ generation according to the absorbance of ABDA at 378 nm in (a-c). **Ce6**: 5 μM ; **Os2** or **Os1**:10 μM ; Light: 633 nm, 6.5 mW/cm^2 .

Ce6 can produce $^1\text{O}_2$ under 633 nm red light irradiation. However, **Os2** cannot generate $^1\text{O}_2$ under 633 nm red light irradiation. When we added both **Os2** and Ce6 in the ABDA solution and then the mixture was irradiated under 633 nm light. The results show that the absorption of ABDA decreases in Ce6 + 633 nm + **Os2** group, which is basically the same as that in Ce6 + 633 nm group, indicating that the addition of **Os2** do not react with (or quench) $^1\text{O}_2$. A similar result occurs with **Os1**.

Supplementary Figure 20. (a, c) The UV-Vis absorption spectra of **Os2** or **Os1** (100 μM) in PBS solution (pH 7.4) containing Ce6 (5 μM) in the dark at 298 K. (b, d) The UV-Vis absorption spectra of **Os2** or **Os1** (100 μM) in PBS solution (pH 7.4) containing Ce6 (5 μM) under light irradiation at 298 K. Light: 633 nm, 6.5 mW/cm^2 .

Ce6 was added to the **Os2** solution and irradiated with 633 nm light, and then the UV-Vis

absorption curves at different irradiation times were measured. The results show that there are no obvious changes in the absorption spectra of **Os2**, meaning that the $^1\text{O}_2$ produced by Ce6 does not react with **Os2**. A similar result occurs with **Os1**.

4) Figure 3, panel a) it is not clear since the arrow pointing to the interpellation with DNA runs, please correct it to make it unambiguous;

Reply to the reviewer: Following the suggestion, we redrew Figure 3a and added in the revised manuscript.

Fig 3a. Schematic diagram of detection of $\text{O}_2^{\bullet-}$, OH^\bullet and $^1\text{O}_2$ with DHE, HPF and SOSG probes, respectively.

5) Phototoxicity in vitro Section - there is no comparison to the cytotoxicity of other preparations currently used on the market in photoactive therapy. Please complete it.

Reply to the reviewer: We thank the reviewer for this suggestion. In the *in vitro* experiments, we used the clinic photosensitizer 5-ALA (5-Amin-levulinic acid) as positive control, and the data have been added to **Table 1**. The dark- and photo- toxicities were investigated in HeLa cells after 8 h 5-ALA drug exposure and 40 h recovery. The results showed that the phototoxicity of 5-ALA ((both $(\text{IC}_{50})_{\text{dark}}$ and $(\text{IC}_{50})_{\text{light}} > 50 \mu\text{M}$) was much lower than that of **Os2** under the same conditions.

Table 1. The dark- and photo- IC_{50} values of the compounds against HeLa cells under normoxia and hypoxia.

Compounds	Condition	Dark [μM]	Light ^[a] [μM]	PI ^[b]
-----------	-----------	------------------------	--	-------------------

Os2	Normoxia (20% O ₂)	89.2±2.6	1.23±0.09	72.5
Os2	Hypoxia (1% O ₂)	>100	5.86±0.19	>18
Os1	Normoxia (20% O ₂)	8.12±0.26	1.31±0.05	6.2
Os1	Hypoxia (1% O ₂)	9.95±0.14	7.51±0.35	1.3
5-ALA	Normoxia (20% O ₂)	>100	68.62±1.99	>1.45
Cisplatin	Normoxia (20% O ₂)	33.22±0.77	47.75±1.5	-

^[a]photoirradiation was imposed (465 nm, 13 mW/cm², 1 h) after 8 h of the complexes incubation and then 40 h recovery; ^[b]photocytotoxicity index, the ratio of (IC₅₀)_{dark}/(IC₅₀)_{Light}. 5-ALA is 5-aminolevulinic acid.

REVIEWERS' COMMENTS

Reviewer #1 (Remarks to the Author):

The authors addressed my critical comments. I can now recommend the revised paper for publication in Nat. Commun.

Reviewer #2 (Remarks to the Author):

The authors corrected and supplemented the manuscript with relevant information. All my remarks have been taken into account. I believe that the revised version of the manuscript can be published in Nature Communications.